# Graphene-Oxide-Grafted Natural Phosphate Support as a Low-Cost Ceramic Membrane for the Removal of Anionic Dyes from Simulated Textile Effluent

**DOI:** 10.3390/membranes13030345

**Published:** 2023-03-17

**Authors:** Hiba Bensalah, Ghizlane Derouich, Xifan Wang, Saad Alami Younssi, Maged F. Bekheet

**Affiliations:** 1Chair of Advanced Ceramic Materials, Institute of Materials Science and Technology, Faculty III Process Sciences, Technische Universität Berlin, Straße des 17. Juni 135, 10623 Berlin, Germany; xifan.wang@ceramics.tu-berlin.de (X.W.); maged.bekheet@ceramics.tu-berlin.de (M.F.B.); 2Laboratory of Membranes, Materials and Environment, Department of Chemistry, Faculty of Sciences & Technics of Mohammedia BP 146, University Hassan II of Casablanca, Casablanca 20650, Morocco; gh.derouich@gmail.com (G.D.); saad.alamiyounssi@fstm.ac.ma (S.A.Y.)

**Keywords:** graphene oxide, low-cost ceramic support, silane grafting, composite membrane, dye removal

## Abstract

A novel natural phosphate/graphene oxide (GO) composite membrane was successfully fabricated using two steps: (i) silane chemical grafting and (ii) dip-coating of a GO solution. First, the low-cost disk ceramic support used in this work was fabricated out of Moroccan natural phosphate, and its properties were thoroughly characterized. The optimized ceramic support was sintered at 1100 °C following a specific heat treatment based on thermogravimetric analysis (TGA) and differential thermal analysis (DTA); it exhibited a permeability of 953.33 L/h·m^2^·bar, a porosity of 24.55%, an average pore size of 2.45 μm and a flexural strength of 22.46 MPa. The morphology analysis using SEM showed that the GO layer was homogenously coated on the crack-free Moroccan phosphate support with a thickness of 2.8 μm. The Fourier transform infrared spectrometer (FT-IR) results showed that modification with silane could improve the interfacial adhesion between the GO membrane and the ceramic support. After coating with GO on the surface, the water permeability was reduced to 31.93 L/h·m^2^·bar (i.e., by a factor of 142). The prepared GO/ceramic composite membrane exhibited good efficiency in the rejection of a toxic azo dye Congo Red (CR) (95.2%) and for a simulated dye effluent (87.6%) under industrial conditions. The multi-cycle filtration tests showed that the rejection rate of CR dye remained almost the same for four cycles. Finally, the flux recovery was also studied. After 1 h of water cleaning, the permeate flux recovered, increased significantly, and then remained stable.

## 1. Introduction

Recently, many filtration processes have been used in order to efficiently remove conventional and emerging pollutants. Ultrafiltration (UF) membrane techniques presented high retention performance for complex macro- and micro-contaminants, such as organic matter, inorganic compounds, hormones and pharmaceuticals [1]. Based on previous research, a consequent connection between membrane retention performance and pollutant properties (molecular size, charge and hydrophobicity) was established [2]. Numerous studies were conducted to enhance the retention performance of ceramic membranes by modifying them with nanomaterials, such as titanium oxides [3] and carbon nanotubes [4], which may decrease the pore size and enhance the charge effects. Graphene, which has a one-atom-thick structure and great mechanical strength and flexibility, has attracted increasing attention since its discovery in 2004 [5]. Because of these distinguishing properties, graphene has the potential to be used in photovoltaic devices, photocatalysis, sensors and medical drugs [6,7]. Due to its chemical inertness, this material is mostly studied in the physics field. Until recently, the potential uses of graphene-based materials in other fields were limited [8]. Molecular simulations, in particular, indicated that graphene-based membranes have a high potential for molecular separation [9].

Because of its exceptional transport properties, excellent chemical stability, and great mechanical strength, graphene oxide (GO) is an emerging material in the area of water purification [10]. Furthermore, its surface and edges consist of abundant functional groups, such as hydroxyl, epoxy and carboxyl groups [11]. These functional groups enable the grafting of GO with other materials to produce a variety of graphene-based compounds [12,13,14,15]. It also shows promise for fabricating water separation membranes due to these hydrophilic groups. To this end, many water purification researchers are attempting to assess the impact of GO nanochannels on different contaminants. Although a few studies reported on the performance of GO membranes, the majority of these have concentrated on evaluating the removal processes of various contaminants in water with polymeric membranes after coating with GO. For instance, Han et al. [16] fabricated a GO membrane based on micro-polyvinylidene fluoride in order to remove organic dyes and inorganic salts from wastewater. The results showed high removal performances of 81% for cationic methyl blue and 99% for anionic direct red. Regarding inorganic salts, it exhibited a maximal retention of 60% for MgSO_4_. Two phenomena can explain these findings: physical sieving and electrostatic interactions. Another study reported the fabrication of a GO membrane on a porous polysulfone substrate [11], showing filtration performances of 95% for Rhodamine WT dye and 46% for Na_2_SO_4_ salt.

While several studies documented the use of polymeric graphene-based mem-branes, the literature on the use of ceramic membranes is lacking. Indeed, the mechanical strength, thermal and chemical stability, and dimensional stability of these supports are the primary barriers to the actual implementation of graphene-based composite membranes. Exploring appropriate supports for developing graphene-based composite membranes is therefore highly encouraged. Compared with polymeric membranes, ceramic membranes are of great interest due to their chemical inertness, porosity structure, thermal and mechanical resistance, and flux performances. Ceramic membranes are usually prepared from metal oxides, such as alumina, silica, zirconia and titania [17,18,19,20,21]. Commercial membranes, especially inorganic membranes, are cost prohibitive, limiting the growth of membrane technologies. One of the challenges for the future growth of inorganic membranes is to produce low-cost membranes from natural non-expensive materials [18,22,23,24,25]. This field already saw rapid growth and innovation. Preparing membranes from low-cost raw materials using basic manufacturing methods would be advantageous in this context, as it would reduce the cost of the ceramic membranes, as well as the energy required by using lower sintering temperatures. Moroccan bentonite [23], pozzolan [26], natural clay [27] and natural phosphate [28] were all used to produce UF membranes for textile effluent elimination or preparation of seawater desalination.

Another major bottleneck for manufacturing efficient composite membranes is the interfacial adhesion between the top active layer and the support surface [29]. Although the bonding strength between the active graphene layer and the membrane support is critical for real applications of graphene-based composite membranes, it has received little attention. As a result, improving interfacial adhesion is required to keep graphene layers from peeling off from the ceramic support.

Accordingly, in this work, we investigated the fabrication of a novel GO/composite ceramic membrane using silane-grafting on a low-cost support made of Moroccan natural phosphate. Chemical grafting was chosen in this work because it was demonstrated to be an efficient approach for improving the bonding strength of composite membranes [30]. 3-Glycidoxypropyltrimethoxysilane (GLYMO) was used for the chemical grafting of the ceramic support in order to improve the bonding strength of the composite membrane. The purpose of this silane modification was to increase the adhesion force between the phosphate support surface and the top graphene oxide layers. During the surface functionalization of the phosphate support, the three methoxyl groups in GLYMO were hydrolyzed and chemically bound to the ceramic support after the thermal treatment. The physical and chemical properties and the morphologies of both the ceramic phosphate support and the GO/phosphate ceramic membranes were systematically investigated via characterization techniques. Ultrafiltration of a toxic azo dye Congo Red (CR) and a simulated dye effluent was also performed in order to evaluate the performance of the new ceramic-supported GO composite membrane for organic dye removal in real separation conditions.

## 2. Materials and Methods

The starting material used for the fabrication of the ceramic support was Moroccan natural phosphate extracted from mine ore from Khouribga, Morocco. The same material was used and extensively characterized as an efficient adsorbent for cationic and anionic dyes in our previous work [31]. The natural phosphate was washed 3 times with distilled water, then dried for 12 h at 100 °C, crushed and sieved at <63 µm. Regarding the synthesis of GO, graphite (KS6) was acquired from (Lonza, Basel, Switzerland), hydrochloric acid (37%) and potassium permanganate were acquired from Sigma-Aldrich (Merck, Berlin, Germany). The silane GLYMO (3-glycidoxypropyltrimethoxysilane) with the chemical formula C_9_H_20_O_5_Si, ethanol 99.8%, sulfuric acid (95–99%) and NaOH pellets were supplied by Sigma Aldrich, Berlin, Germany. The solution of simulated dye effluent was prepared by dissolving a given quantity of NaCl salt in analytical grade Congo Red (dye content ≥ 35%) solution (Sigma-Aldrich, Berlin, Germany).

### 2.1. Characterization Techniques

The thermal behavior of the natural phosphate was investigated using thermogravimetric analysis (TGA) and differential thermal analysis (DTA) in air with a heating rate of 10 °C/min from 20 °C to 1100 °C (NETZSCH STA409PC, Selb, Germany). Prepared membranes were subjected to many characterizations. The porosity was measured in accordance with ASTM C373. The flexural strengths of the fabricated ceramic support according to the different sintering temperatures and of the GO/composite membranes at different GO concentrations were measured in a three-point bending load using 1 kN Screw Type Flat Grips provided by Shimadzu (Kyoto, Japan). The surface and cross-section morphologies of the raw ceramic support and GO membrane were investigated using scanning electron microscopy (SEM) on a Gemini Leo 1530 (Zeiss, Göttingen, Germany). The fixtures conformed to ASTM C393, C1161, D790 and related standards. The functional groups of the GO powder, GLYMO, silane-modified support and GO/ceramic (phosphate) composite membrane were measured via the attenuated total reflection (ATR) method using Fourier transform infrared spectrometer FT-IR (Bruker EQUINOX 55, Billerica, MA, USA). Turbidity was tested using a mobile turbidimeter (HACH 2100Q, Loveland, CO, USA).

### 2.2. Synthesis of GO

GO was synthesized in our laboratory via the oxidation of graphite using the improved Hummer’s method and was already extensively reported [32]. Briefly, a mixture of concentrated H_2_SO_4_ and HNO_3_ (ratio 9:1) was incorporated, followed by magnetic stirring of graphite flakes. Afterward, 18 g of KMNO_4_ was subsequently slowly added to the reaction to avoid overheating. The reaction mixture was then heated to 50 °C in a water bath for 12 h, then cooled to room temperature, and diluted by gradually dropping the reaction mixture onto ice. The yellow suspension confirmed that the graphite was oxidized. The resulting suspension was dispersed in a 15 vol.% aqueous HCl solution, followed by centrifuging and washing. Lastly, the GO that resulted from this multiple-wash procedure was dispersed in ethanol and vacuum dried at 55 °C, yielding 0.97 g of GO powder from the initial 5 g graphite (yield 30.2%).

### 2.3. Fabrication of Ceramic Supports

The flat ceramic support was made of raw Moroccan natural phosphate without adding any chemical additives. Phosphate powder was uniaxially pressed in a cylindrical stainless steel mold at 15 MPa for 15 min using a hydraulic press to produce flat disks with a diameter of 40 mm and a thickness of 2 ± 0.2 mm. The green compacted supports were later sintered in a muffle furnace (Nabertherm L9/13/P320, Lilienthal, Germany) following a specific heat treatment based on thermal analysis of the natural phosphate up to 1000 °C, as shown in Figure 1a. The temperature increased gradually at 2 °C/min, then 3 °C/min from room temperature to 1000 °C. The low heating rate of 2 °C/min in the first stage was used in order to prevent cracks from forming in the phosphate ceramic support. The second heating rate of 3 °C/min until 1000 °C was adopted for the consolidation of the ceramic membrane. Shrinkage (%) was calculated by measuring the diameter of the support before (D_0_) and after (D_1_) sintering:
(1)
$$\mathrm{Shrinkage}=\frac{D_{0}-D_{1}}{D_{0}}\times 100$$


As shown in Figure 1b, the continuous increase in sintering temperature increased the percentage of mass shrinkage, reaching 13.8% at 1000 °C (i.e., the diameter of the green body shrunk from 40 mm to 34.5 mm in the sintered support). This shrinkage can be explained by the burning of organic components and the decomposition of carbonate groups. The results of the heat treatment used in this work agreed with previous studies [28].

The porosity of the fabricated membrane as a function of sintering temperature is presented in Figure 1c. It can be noticed that the porosity decreased from 32.34 to 24.55% when the sintering temperature increased from 900 to 1100 °C. This might be explained by the increased connectivity between the phosphate particles during calcination [33]. Beyond the sintering temperature of 1050 °C, the membrane porosity decreased drastically. This could have been due to both an increase in the degree of densification and the prevalence of the vitrification phenomenon. As a result, the thermal treatment process caused both an increase in density and a decrease in porosity. This result is consistent with the previously observed porosity evolution as a function of sintering temperature [25].

### 2.4. Surface Modification of the Ceramic Membrane

Before coating, the ceramic support surface was polished with a SiC abrasive paper, cleaned with sonication in distilled water for 10 min and then dried for 12 h at 100 °C. To coat the GO layer on the surface of obtained phosphate supports, the supports were chemically functionalized first with silane GLYMO precursor, followed by dip-coating in GO solution, as shown in Figure 2. For the surface functionalization of phosphate support, the KSV NIMA Dip Coater was used for dip-coating of the membrane surface in a solution of GLYMO/ethanol (2 mg/mL) with a constant rate of emergence (200 mm/min) at 40 °C for 40 min. Then, the functionalized ceramic support was dried at 100 °C for 5 h.

After GLYMO coating, the GO layer was dip-coated on the support surface using different concentrations of aqueous GO solution (i.e., 0.1, 0.5, 1, 2, 4 and 6 mg/mL). To prepare the aqueous GO solution, a given amount of GO powder was dispersed ultrasonically in 50 mL deionized water for 2 h to prepare a homogenous suspension of GO nanosheets in the solution [34]. As shown in Figure 2b, the obtained GO suspension without an ultrasonic treatment was heterogeneous, while the use of an ultrasonic treatment led to a highly stable suspension of GO solution for more than one month. The successful coating of GO layers on the support membrane was confirmed by the change in the color of the membrane, as shown in Figure 2c. After the GO coating process, the obtained ceramic membrane was dried at 50 °C for 28 days in order to enhance the bond strength of dip-coated GO on the membrane.

It was observed that the dilute concentrations of the GO solution below 2 mg/mL did not adhere well to the silane-grafted ceramic support, and many mechanical defects, such as macro-cracks, were noticed on the surface of the membranes (Figure 2c). Quite the opposite was noticed when using higher concentrations of GO solution (4 mg/mL and 6 mg/mL), where GO sheets were easily peeled off from the too-thick GO membrane. Accordingly, we chose 2 mg/mL as a suitable concentration to fabricate homogenous and stable GO/ceramic composite membranes.

### 2.5. Filtration Test

Water permeability experiment on the natural phosphate support was conducted from 0 to 0.12 bar using distilled water at 25 °C ± 2 °C for 2 h on a home-made apparatus made of glass, as illustrated in Figure 3a. The raw ceramic support was placed in a vertical position and perpendicular to the water flow direction in a membrane cell with an effective area of 5.3 cm^2^. The vertical pressure variation technique was used, which is described by Equation (2):
(2)
ΔP=−ρgΔh

where P is the pressure, 
ρ
 is the density, g is the acceleration due to gravity, and h is the height.

The prepared GO/ceramic composite membrane was used in order to treat a Congo Red dye solution. The filtration experiments were conducted in a UF pilot plant mounted in the laboratory, as shown in Figure 3b. Briefly, the UF installation of the cross-flow system consisted of the following major parts: a ceramic UF module, an adjustable circulation pump, a 3 L thermostatic feed tank, pressure gauges, a valve that was used to control the pressure in the system and a permeate sampling beaker. The flow rate was controlled using a flow meter, and the flow pressure before and after the membrane was monitored using two manometers. The UF experiments were performed by applying a transmembrane pressure ΔP (bar) that varied from 0 to 3 bar. After reaching constant flow conditions, the permeate volume fluxes and rejection factors were calculated for the experimental dye. During the UF experiments, the temperature of the flowing solutions was kept constant (at 25 ± 2 °C) and repeated three times. After each experimental run, the ceramic modules were cleaned under dynamic conditions with the use of a 0.2 N NaOH solution.

The permeate flux J (L/h·m^2^) and water permeability L_P_ (L/h·m^2^·bar) were determined using Equations (3) and (4):
(3)
$$J=\frac{V}{A\times t}$$


(4)
$$J=L_{P}\times \Delta P$$

where V (L) is the permeate volume, A (m^2^) is the effective membrane area during filtration time t (h) and ΔP (bar) is the transmembrane pressure.

The prepared membrane was evaluated via filtration of the CR aqueous solutions with an initial concentration of 20 ppm under a constant transmembrane pressure of 3 bar for 2 h. The permeate was collected every 20 min during filtration for characterization. To determine the concentration of the dye effluent, UV-vis spectroscopy was used. The dye rejection R (%) was determined by measuring the initial and final concentrations and then using Equation (5):
(5)
$$R=\left(1-\frac{C_{i}}{C_{f}}\right)\times 100$$

where C*_i_* and C*_f_* are the dye concentrations (ppm) in the feed and permeate, respectively.

The antifouling tests of the GO/phosphate membrane were carried out at 25 °C and a constant transmembrane pressure of 3 bar for 2 h. The antifouling parameters, namely, the flux recovery ratio (*FRR*), total fouling ratio (*TFR*), reversible fouling ratio (*RFR*) and irreversible fouling ratio (*IFR*), were calculated using Equations (6), (7), (8) and (9), respectively:
(6)
$$FRR\;\left(\%\right)=\frac{J_{w0}}{J_{w1}}\times 100$$


(7)
$$TFR\;\left(\%\right)=\left(1-\frac{J_{p}}{J_{w0}}\right)\times 100$$


(8)
$$RFR\;\left(\%\right)=\frac{J_{w1}-J_{p}}{J_{w0}}\times 100$$


(9)
$$IFR\;\left(\%\right)=\frac{J_{w0}-J_{w1}}{J_{w0}}\times 100$$

where *J_w_*_0_ (L/h·m^2^) is the water permeate flux of the GO/phosphate membrane, *J_p_* (L/h·m^2^) is the permeate flux of CR at 100 ppm (dye concentration), and *J_w_*_1_ (L/h·m^2^) is the water permeate flux after cleaning the GO/phosphate membrane distilled water for 1 h using the UF pilot at 3 bar (transmembrane pressure).

In order to optimize the GO/phosphate membrane in commercial applications, the produced membrane was used to filter the simulated dye effluent, which was made by adding 1 g/L of NaCl to 100 ppm of CR dye solution. The rejection experiment was performed at a transmembrane pressure of 3 bar, an initial pH of 6 and room temperature.

## 3. Results and Discussion

### 3.1. Thermal Behavior of the Moroccan Natural Phosphate

In the first step, we determined the optimal sintering conditions of the Moroccan natural phosphate by investigating the thermal behavior of the raw material using TGA/DTA from room temperature to 1100 °C. As shown in Figure 4, a first weight loss of 0.64% was observed from the TGA curve of the phosphate between 25 °C and 200 °C, which was supported by an endothermic peak at ~160 °C in the DTA curve, attributing to the desorption of water. A second mass loss of 1.78% between 200 °C and 460 °C, along with an exothermic peak at ~360 °C in the DTA curve, was due to the elimination of organic impurities in the raw phosphate material [35]. The last weight loss of 5.56% between 450 °C and 900 °C, with an exothermic peak at ~700 °C, was attributed to the decomposition of mineral carbonates present in the phosphate. No further significant weight loss was observed above 1000 °C; thus, this temperature was chosen for the sintering of the phosphate ceramic support in this work.

### 3.2. GO Properties

In order to confirm the synthesis of GO, XRD and FT-IR analysis were performed, as shown in Figure 5. In Figure 5a, the diffraction peak observed at 2θ ~ 11°is attributed to the (001) plane of GO, showing that the graphite was completely oxidized to GO. The XRD pattern shows that the GO nanoparticles consisted of a single crystalline phase, and no impurity phase existed.

The FT-IR spectrum of the GO powder was recorded between 400 and 4000 cm^−1^, as presented in Figure 5b. The large and high absorption peak at 3437 cm^−1^ referred to the –OH stretching vibration, accompanied by the adsorption of water molecules existing due to the humid environment when GO was kept and ground. The high surface area of the material may have been the cause. The absorption band at 1735 cm^−1^ corresponded to C=O from carbonyl or conjugated carbonyl groups. The presence of an absorption peak at 1626.4 cm^−1^ was designated to the stretching vibration of C=C. The absorption peaks at 1401 cm^−1^, 1214 cm^−1^ and 1102 cm^−1^ were attributed to the C–OH unit from carboxyl group, C–O–C from epoxy or ether, and C–O from alkoxy, respectively. These findings are in good agreement with the literature [36]. These GO functional groups may enhance the bonding between the GO layer and silane via active sites.

### 3.3. Flexural Strength

The evolution of the mechanical strength of the ceramic support as a function of the sintering temperature and of the prepared GO/ceramic membrane as a function of the GO concentration is presented in Figure 6. In Figure 6a, the flexural strength of the ceramic support increased from 14.25 to 22.46 MPa during the sintering process from 900 to 1100 °C. The densification effect can explain the increase in mechanical strength: as sintering temperatures rise, phosphate particles become more firmly connected, forming stronger ceramics [37]. The peak value of flexural strength was found to be 26.15 MPa at a GO solution concentration of C_GO_ = 2 mg/mL, with a 16.43% mechanical improvement compared with the ceramic support before coating (Figure 6b). This can be explained by GO sheets being uniformly coated on the supports surface, in addition to the excellent mechanical characteristics of GO and the hydrogen bonds formed by the several oxygen-containing active sites on its surface, which increased the intermolecular interactions, thus enhancing the flexural strength. As a result, the fabricated GO/ceramic composite membrane could readily sustain the hydraulic pressure used in the UF experiments.

### 3.4. Morphology and Surface Analysis

In order to demonstrate the successful graft of GLYMO on this phosphate support and the elaboration of the GO/ceramic membrane, FT-IR characterization was applied. Figure 7a shows the characteristic absorption bands of the silane material, such as the −CH_2_− bands at 2872 cm^−1^ and 2950 cm^−1^. Meanwhile, the peaks at 1100 cm^−1^ and 1195 cm^−1^ refer to the epoxy group. In addition, the peak at 1000 cm^−1^ corresponded to the Si−O−C bond. As shown in Figure 7c, these characteristic bands were observed in the FTIR spectra of the surface coating of the phosphate support with GLYMO, which was absent in the ceramic support without modification (Figure 7b). These results confirmed the successful coating of the silane layer onto the surface of the phosphate support [38]. The successful coating of the GO layer on the support surface was also confirmed by detecting the characteristic absorption bands of GO at 1102 cm^−1^ for the alkoxy group C−O, 1214 cm^−1^ for the epoxy group C−O−C and 1735 cm^−1^ for C=O in the FT-IR spectra of the GO/phosphate membrane (Figure 7d). The silane used in this work, namely GLYMO, is a short-chain molecular link silane with few methoxy groups that can be covalently bonded on the phosphate surface. During the dip-coating step, the pH of the GO solution was kept at under 6 in order to create a reaction between the GO and the epoxy group of GLYMO. Indeed, it was reported that GO sheets tend to graft to GLYMOs epoxy group via a ring-opening reaction under an acidic environment [39]. Thus, strong adsorption of GO on the modified ceramic support was achieved, leading to a stable interfacial adhesion between the GO layer and the support surface.

The surface and cross-section morphology of the phosphate ceramic support were observed using SEM (Figure 8). The morphology of the ceramic support surface after calcination is shown in Figure 8a. The surface was smooth, and no significant macro-defects (cracks) were observed. Figure 8b exhibits the SEM image of the cross-section of the phosphate ceramic porous support. It can be noticed that the porosity was fairly small, with an average pore size of 2.45 μm and the ceramic support was asymmetric.

SEM images (Figure 9) show the microstructures of the phosphate support before and after coating with the GO layer. Figure 9b shows the uniform coating layer of the GO on the phosphate support. No cracks, damage, or detachment were observed on the surface of the membrane due to strong hydrogen bonding within the GO coating. The GO sheets seemed to be linked to each other in Figure 9c and formed a thin layer on the silane-modified ceramic supports. The cross-section image (Figure 9d) reveals that the GO coating of approximately 2.8 μm adhered well to the ceramic support. These results indicate the successful GO coating layer on the silane-treated phosphate support using the dip-coating method, confirming the FT-IR findings. The relevant adsorption of GO on the ceramic support surface was due to the capillary force of the GO sheets piled layer by layer during the dip-coating step. The water in the GO solution infiltrated into the pores of the phosphate ceramic support. It was reported that GO sheets are spread in the same direction on the support surface due to the liquid crystal characteristic of GO [40]. Consequently, the GO layer was coated automatically onto the surface of the support. When GO sheets come in contact with a silane-modified ceramic support, polymerization between GO and epoxy groups from GLYMO will occur, resulting in a high bonding interaction between the phosphate support and the GO layer.

### 3.5. Phosphate Support Performance

The performance of the phosphate ceramic support was evaluated by studying several parameters, including the chemical resistance, water permeability and selectivity toward particles. Figure 10a presents the corrosion resistance test results of the elaborated support. The test was carried out by measuring the mass loss in acidic (H_2_SO_4_ solution at 0.1 M) and basic media (NaOH solution at 0.1 M) for 6 days at 80 °C. The results show that the mass loss of the phosphate support did not exceed 0.8 wt.% in the acidic medium and 3.2 wt.% in the basic medium, suggesting a better chemical resistance of the phosphate support in the acidic medium than in the basic medium. To determine the filtration type of the support, the water permeability was studied. Figure 10b displays the permeate flux of water as a function of time. The water permeability value determined from the linear equation of each line was about 953.33 L/h·m^2^·bar, suggesting the microfiltration (MF) type for the bare support. Additionally, the selectivity of the phosphate support was investigated via turbidity removal using baking powder with an initial turbidity of 500 NTU. According to Figure 10c and from the first 5 min of filtration, the phosphate support could effectively remove the suspended matter and colloidal particles from the solution under study with a percentage of 99.9% after 120 min.

### 3.6. GO/Phosphate Composite Membrane Performance

The evaluation of the water flux was the first filtration test that needed to be studied to determine the water permeability of the GO/phosphate membrane. The water flux was investigated after filtration for 20 min (stabilization time). Figure 11a shows the water flux as a function of the transmembrane pressure. The permeability of the modified membrane was about 31.93 L/h·m^2^·bar, equivalent to 97.7 L/h·m^2^ as a water permeate flux at a transmembrane pressure of 3 bar. By comparing the results of the bar phosphate support and GO/phosphate membrane, it was found that the GO layer reduced the water permeability by a factor of 142. Figure 11b presents the CR rejection percentage as a function of time with different CR concentrations from 20 to 100 ppm at an initial pH of 6 and a transmembrane pressure of 3 bar. It can be observed that the rejection percentage of the CR dye was high. This means that the hydrophilicity of the GO/ceramic membrane was confirmed. This may have been due to hydrogen bonds between the water and the hydroxyl and carboxyl groups existing on the surface of the GO membrane. Accordingly, the water molecules could easily and quickly permeate through the interspace between the brick-like GO layers.

The CR rejection rate weakly increased from 83.4 to 95.2% when the dye concentrations and filtration time increased. This may have been due to the accumulation of retained dye molecules on the membrane surface, leading to the existence of a concentration polarization layer that led to the diffusion resistance of the molecules of dyes because of electrostatic repulsion between the both negatively charged polarization layer and dyes in the solution. Indeed, at the given pH = 6, and according to the literature, the point of zero charge of GO was observed to be pHpzc = 2.4. Similar observations were reported [12]. This pHpzc suggests that the surface of GO should be predominantly positive at pH values lower than 2.4 and negative at pH values higher than 2.4. Some suggestions can be made in order to tackle the effect of concentration polarization, for instance, controlling the hydrodynamic operating parameters, such as the pressure and flow speed.

### 3.7. Antifouling Performance of the GO/Phosphate Membrane

Antifouling performance is a crucial sign of GO/phosphate membrane long-term stability and application at a large scale [41]. Figure 12a shows the permeate flux of pure water, CR solution and pure water after the physical cleaning of the membrane. According to the results, it can be seen that the permeate flux was significantly reduced due to the cake layer formation on the membrane surface that resulted from the accumulation of retained CR dye molecules; this process is called concentration polarization and fouling of the membrane [42]. After cleaning the surface of the membrane with pure water for 1 h, the permeate flux recovered, increased significantly, and remained stable.

As observed in Figure 12b, the flux recovery ratio (FRR) value was 76.91% and the corresponding total fouling ratio (TFR) value was 38.67%. This revealed that the GO/phosphate membrane showed a higher flux recovery and lower total fouling ratio. However, the values of the reversible (RFR) and irreversible (IFR) fouling ratios were equal to 15.58 and 23.09%, respectively. These rates indicated that 15.58% of the fouling can be removed via simple hydraulic cleaning, while 23.09% of the pore blockage was irreversible and could not be removed, except via chemical cleaning [43].

### 3.8. Recyclability Studies of GO/Phosphate Membrane

The recyclability studies were repeated for four cycles using the same membrane after physical cleaning with distilled water and rinsing the GO/phosphate membrane for 1 h using the UF pilot at a transmembrane pressure of 3 bar. The CR solution used a concentration = 100 ppm, where the rejection rate reached its maximum value (59.92 L/h·m^2^). The results (Figure 13a) show that the permeate flux slightly decreased (from 59.92 to 54.27 L/h·m^2^) after each cycle, which may have been due to cake formation on the membrane surface and pore blockage by CR molecules [44]. As shown in Figure 13b, the rejection rate of CR dye remained almost the same for all four cycles (varied between 95.2 and 94.9%), which shows that the GO/phosphate membrane can be used repeatedly and safely for CR dye filtration.

### 3.9. Rejection of Simulated Dye Effluent

The lack of studies on large-scale applications of GO/composite ceramic membranes, as well as the fact that the majority of the reviewed literature involves synthetic effluents synthesized in the laboratory, make it difficult to understand the processes involved in real-world applications. The existence of organic and inorganic compounds in industrial effluents can inhibit or change the efficiency of these membranes.

Accordingly, the GO/phosphate membrane was evaluated using a CR dye and NaCl salt mixture solution to investigate its performance in the rejection of real polluted wastewater samples. As shown in Figure 14, the permeate flux and rejection rate decreased in the presence of NaCl in the CR dye solution. In the absence of NaCl, the permeate flux and rejection rate of the pure CR solutions (at 3 bar) reached a value of 59.92 L/h.m^2^.bar and 95.2%, respectively, while for the CR-NaCl mixture, the value decreased by 11% to achieve 53.32 L/h.m^2^.bar for the permeate flux and by 8% to achieve a rate of 87.6% for the CR-NaCl rejection. This decrease in the permeate flux and rejection rate in the presence of NaCl salt in CR dye solution can be explained by the shield of GO/phosphate membrane charge and the greater dispersion of CR dye [45]. These effects might facilitate the accumulation of CR dye molecules on the GO/phosphate membrane surface, blocking the membrane’s pores and, consequently, decreasing the permeate flux and rejection rate of the solution.

To emphasize the novelty of this study, a thorough comparison of the membrane separation efficiency with previously reported data is provided in Table 1.

This comparison was investigated based on newly reported data in the literature with regard to the permeation water permeability (PWP) and dye removal efficiency (%R) of GO composite membranes. Relatively few ceramic supports coated with GO layer were recently reported for UF applications. From Table 1, it can be distinguished that most of the GO composite membranes reported used polymeric/commercial supports. Compared with our work, on top of using a low-cost ceramic support and a facile one-step grafting method, our fabricated GO/membrane showed competitive rejection of dyes.

## 4. Conclusions

GO sheets were synthesized from pure graphite via the modified Hummer method and were successfully coated onto a low-cost ceramic membrane using silane grafting via one-step dip-coating at room temperature. This technique was straightforward since it employed automatic capillary infiltration and did not require any external pressure. It was confirmed that the modification with silane (GLYMO) improved the interfacial adhesion between the GO layer and the phosphate ceramic support. The GO/ceramic composite membrane was stable and homogeneous and no defects could be noticed. The GO/ceramic phosphate membrane exhibited high rejection for the anionic azo dye CR and under industrial conditions of simulated dye effluent. The rejection rate of CR dye remained almost the same for four cycles, which could permit long-term usage of the membrane. Accordingly, this work suggests a simple two-step strategy in order to fabricate stable GO membranes on low-cost ceramic supports with great interfacial adhesion, potentially contributing to large applications of graphene-based membranes.

Regeneration approaches are crucial to company operations in an industrial setup because they evaluate the cost efficiency and the life cycle of membranes. Nevertheless, only a few publications provided recyclability studies, which may greatly reduce interest in the industrial sector. As a result of our work’s interesting findings, further studies must take particular note to conduct their analysis in mixed contaminants effluents and upscaling the membrane feasibility from laboratory conditions to an industrial pilot.

## Figures and Tables

**Figure 1 membranes-13-00345-f001:**
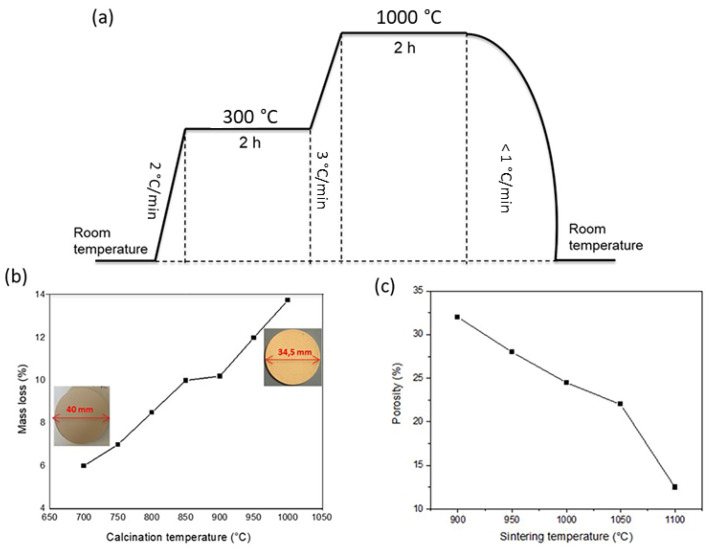
(**a**) Scheme of the thermal treatment program; (**b**) mass loss as a function of calcination temperature of the ceramic support made of natural phosphate; (**c**) variation of phosphate ceramic support porosity vs. sintering temperature.

**Figure 2 membranes-13-00345-f002:**
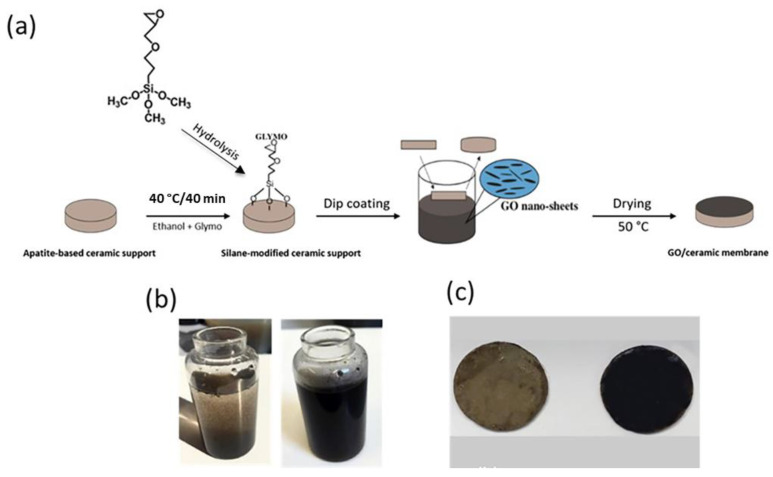
(**a**) Steps of the preparation of the GO/ceramic (phosphate) modified membrane; (**b**) the heterogeneous graphene oxide solution was heterogeneous without an ultrasound treatment (**left**) and was homogeneous and stable with an ultrasound treatment (**right**); (**c**) image of the ceramic support with silane-grafting after dip-coating using a GO solution of CGO = 0.5 mg/mL (**left**) and CGO = 2 mg/mL (**right**).

**Figure 3 membranes-13-00345-f003:**
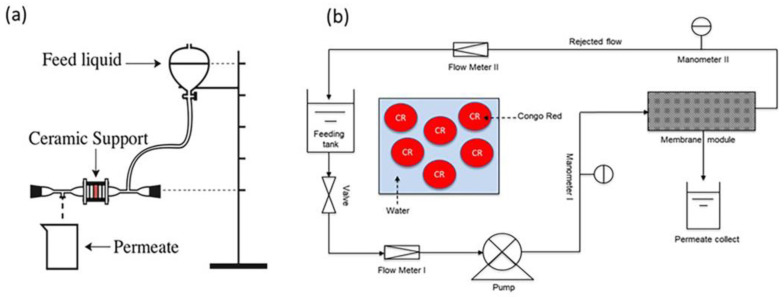
(**a**) Scheme of the laboratory apparatus for the permeability test; (**b**) schematic for the CR separation system with the GO/ceramic membrane.

**Figure 4 membranes-13-00345-f004:**
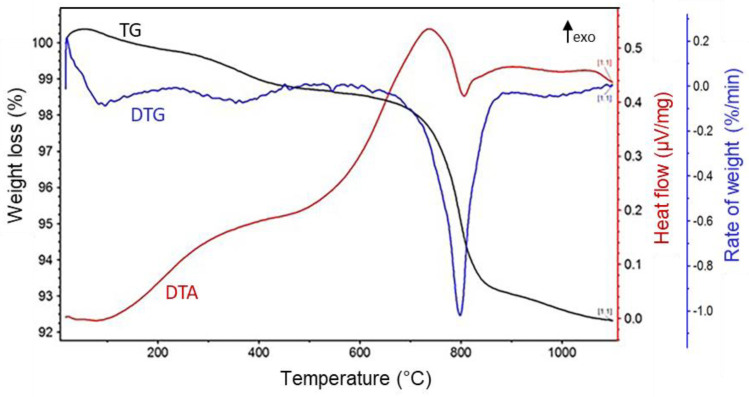
TGA and DTA curves of the Moroccan natural phosphate used for ceramic supports.

**Figure 5 membranes-13-00345-f005:**
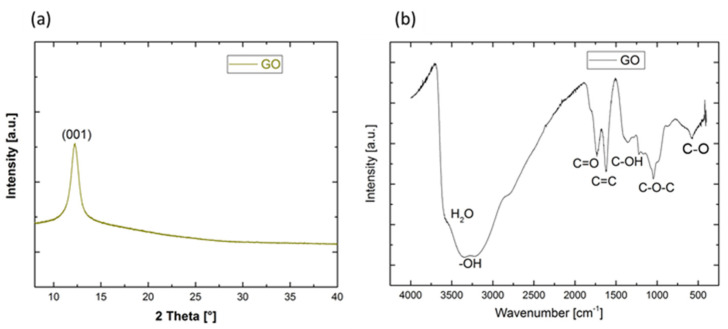
(**a**) XRD pattern and (**b**) FTIR spectra of the prepared GO.

**Figure 6 membranes-13-00345-f006:**
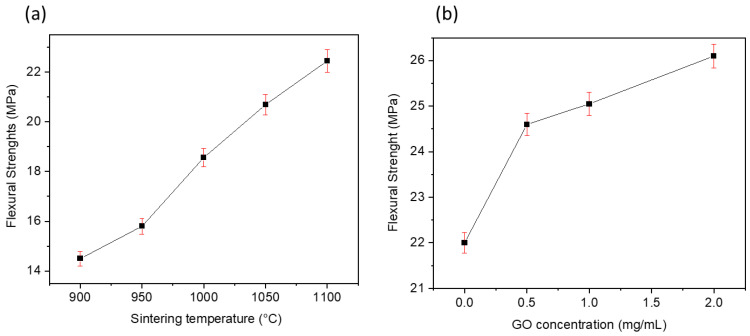
(**a**) Effect of calcination on the flexural strength of the phosphate ceramic support; (**b**) effect of the GO concentration on the flexural strength of the GO/ceramic membrane.

**Figure 7 membranes-13-00345-f007:**
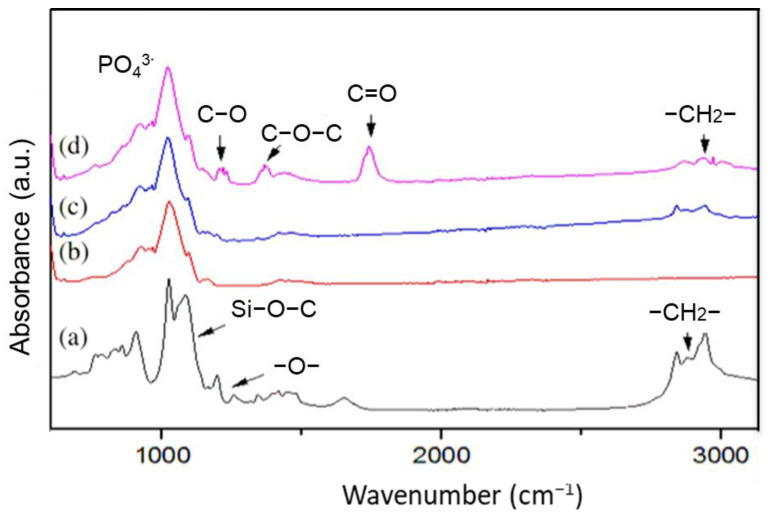
FT-IR spectra of (**a**) GLYMO, (**b**) the ceramic support made of natural phosphate, (**c**) the modified ceramic support grafted using silane, and (**d**) the GO/ceramic membrane.

**Figure 8 membranes-13-00345-f008:**
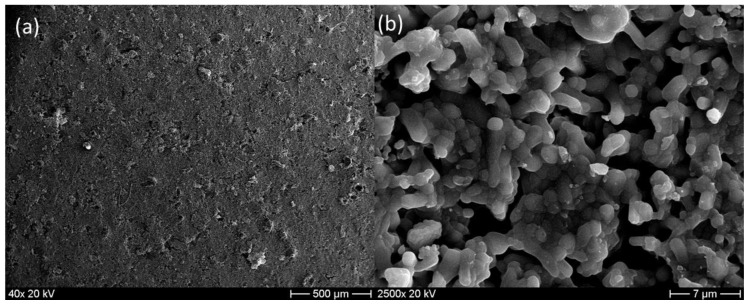
SEM images of the phosphate ceramic support (**a**) surface and (**b**) cross-section view.

**Figure 9 membranes-13-00345-f009:**
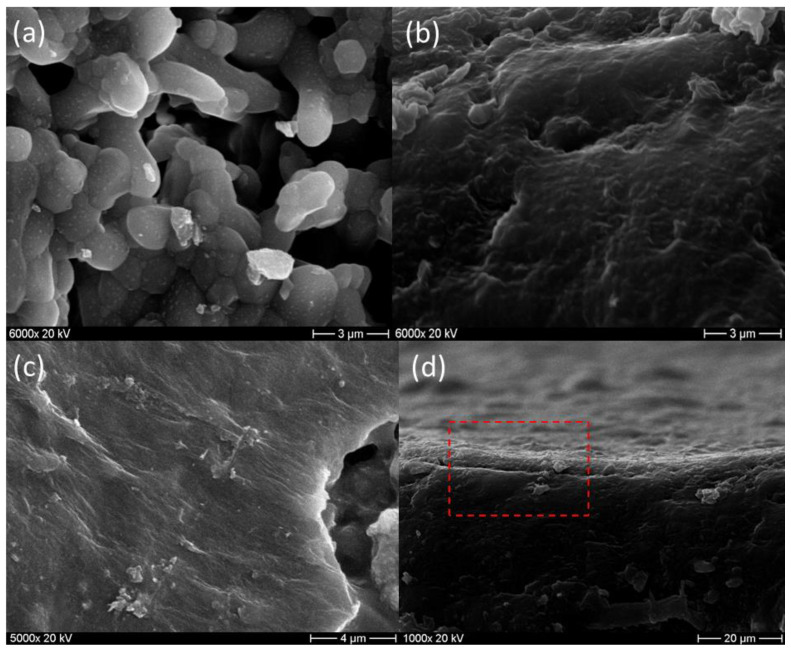
SEM images of (**a**) the phosphate ceramic support before coating with GO, (**b**) the GO/ceramic composite membrane surface after coating, (**c**) the GO layers coated on the membrane surface and (**d**) the cross-section of the GO/ceramic composite membrane.

**Figure 10 membranes-13-00345-f010:**
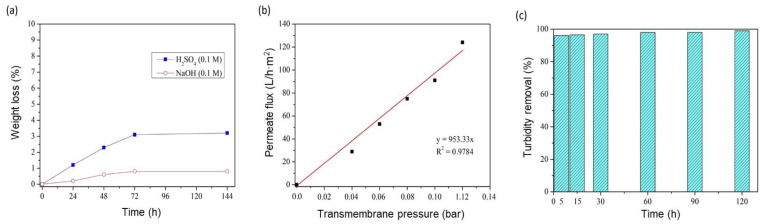
(**a**) Weight loss of the phosphate support in acidic and basic media; (**b**) variation of the water permeate flux of the phosphate support as a function of the transmembrane pressure; (**c**) turbidity removal of the phosphate support as a function of time.

**Figure 11 membranes-13-00345-f011:**
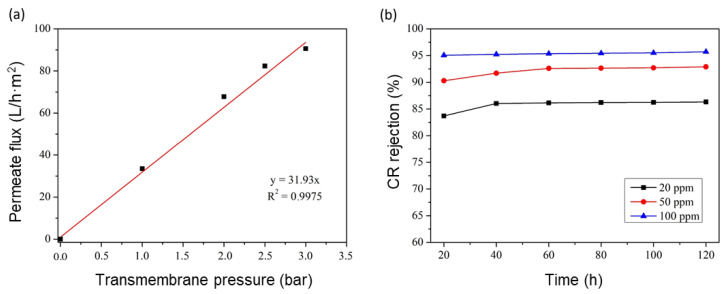
(**a**) Variation in water permeate flux of the GO/phosphate membrane as a function of the transmembrane pressure; (**b**) CR rejection rate as a function of the concentration.

**Figure 12 membranes-13-00345-f012:**
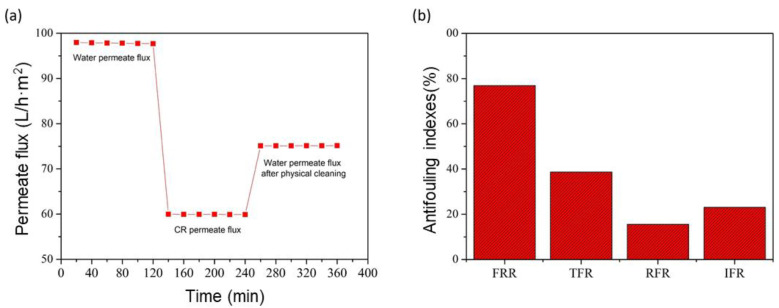
(**a**) Permeate flux during the filtration experiments; (**b**) antifouling indexes of the GO/phosphate membrane.

**Figure 13 membranes-13-00345-f013:**
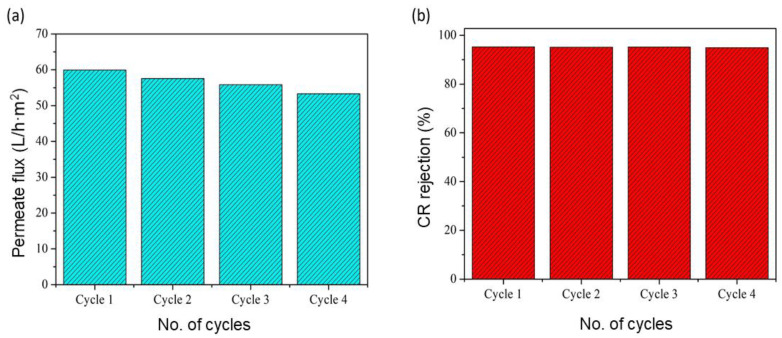
Recyclability studies of GO/phosphate membrane: (**a**) permeate flux and (**b**) rejection rate of CR dye for four cycles.

**Figure 14 membranes-13-00345-f014:**
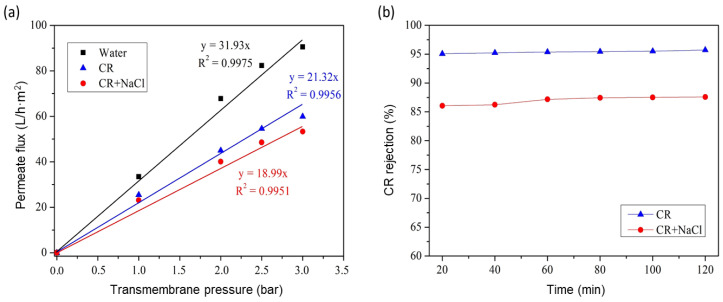
(**a**) Variation of the water permeate flux; (**b**) rejection rate of the pure CR dye and CR-NaCl mixture.

**Table 1 membranes-13-00345-t001:** Overview of recent publications on GO/composite membranes for dye rejection.

Composite Membrane	Modification Method	PWP (L/h·m^2^·bar)	Dyes	Removal Efficiency (%)	References
GO-PCS	Surface coating	NA	Direct black (100 mg/L)	99.80	[46]
Cu(tpa)@GO/PES	Phase inversion	0.7867	Methyl blue Methyl orange Congo red	15 65 90	[47]
rGO/Pozzolan	Spin coating	2	Bromothymol blue Methyl orange Murexide	94 93 97	[26]
GO/commercial ZrO_2_-TiO_2_	Vacuum filtration	5.1	Acid Orange 7 Reactive Black 5 Direct Blue 71	99 96 92	[48]
MPES	Pressure-assisted filtration	99.40	Blue Corazol	97.80	[49]
GO/PES	Non-solvent-induced phase separation	13	Sunset yellow Acridine orange	62.3 35.4	[50]
Sulfonated (s-GO)/PES	Phase inversion	9.1	Acid blue Bismark Brown	83.9 83.5	[51]
GO/cellulose acetate	Vacuum-assisted filtration	NA	Methylene blue	99	[52]
GO/natural phosphate	Silane grafting + dip coating	31.93	Congo red	95.2	This work

Abbreviations: PES—polyethersulfone; Cu(tpa)@GO—metal–organic framework of copper, terephthalate acid (tpa) and GO; MPES—microfiltration membranes of polyethersulfone; PCS—phosphorylated chitosan; PWP—pure water permeability.

## Data Availability

Not applicable.

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
