# Peer review of "Graphene-Oxide-Grafted Natural Phosphate Support as a Low-Cost Ceramic Membrane for the Removal of Anionic Dyes from Simulated Textile Effluent"

_membranes, 2023, doi:10.3390/membranes13030345_

Round 1

Reviewer 1 Report

The author could integrate GO membranes on low-cost ceramic supports. The manuscript is well written. 

I recommend major revision based on the following comments. 

- Why did you choose phosphate as a basis for ceramic supports?  

- Please, state the instrument used for pressuring the powder. 

- Please, refer the FT-IR bands to relevant recent publications. For instance, https://doi.org/10.1186/s42834-022-00131-0

- What is the stability of the prepared membranes with re-using?  

- Figure 7c, please add the turbidity removal with time less than 30 min on the graph.

- Please, illustrate the anti-fouling effect. 

- Please, rewrite the conclusion section with further significant findings and recommendations. 

Reviewer 2 Report

1-        The first part of the abstract looks like an introduction, please revised, and contain the results obtained from the research into this section.

2-        Why only anionic dyes and not cationic??

3-        The SEM images does not really shows any nanosheets, even the SEM scale is 10 mm, please revised.

4-        There are many important characterization required for this publications such as XRD and the mechanical properties such as tensile strength, of the prepared membrane.

5-        Please study the applications of the low-cost ceramic membrane for the removal of   for the removal of anionic dyes from real polluted samples.

6-        Also, please study the possible interference with other cationic dyes or high concentration salts such as NaCl.

7-        Could the others provide SEM analysis for the membrane after the cycle stability?

After a clear thought, I may recommend this manuscript for publication after careful consideration of the above mention comments,

Reviewer 3 Report

This is interesting research, but the following minor problems should be addressed.

1.      Affiliations next to authors' names - is a), b), should be 1), 2)

2.      Line 46, order of citation of literature

3.      Specify how many times it was washed and how many hours of dried phosphate?

4.      What was the working atmosphere in the furnace during heat treatment? What was the pressure like?

5.      Specify how many hours the ceramic was dried before the coating was applied?

6.      Line 219 the unit of area is cm2

7.      In formula (4) there is a "delta t" factor, and the description of the formula is "t"

8.      Poor quality and legibility figure 4

9.      Line 271 - can the FT-IR spectrum (Figure 5c) be used to evaluate membrane surface defects?

10.   Poor picture quality 5

11.   Please give reasons for this statement: „Also, it can be shown that the most important FTIR band of the natural phosphate PO43- is also present at 1020 cm-1, confirming the preparation of a strong and stable GO/ceramic composite membrane.”

12.   To determine how strong the adhesion between GO and the substrate is, it would be necessary to conduct adhesion tests

13.   Needs clarification - Can you conclude from the comparison of the FTIR spectra that there is a chemical bond between the phosphate support and the GO coating? To confirm the formation of this type of bond, it was necessary to perform FTIR of an additional sample - phosphate support with GO coating without silane treatment

14.   Does good adhesion of GO to the substrate not result from the development of the surface of the phosphate support?

15.   For a better assessment of the adhesion of the GO layer to the substrate, an SEM image should be taken on the GO/ceramic section at a higher magnification than shown in Figure 6c.

16.   Is the GO layer present only in places marked with an arrow in Figure 6b? Are these arrows needed in this photo?

17.   Literature prepared not in accordance with the requirements of the Membranes journal

18.   Poor quality of figures 7 and 8

19.   The comparison in Table 1 should only refer to tests performed on the same type of dye.

20.   I think the results should be separated from the discussion for better article readability.

21.   The paragraph (lines 383 to 395) should be moved to Conclusions.

Round 2

Reviewer 1 Report

The authors have addressed all points and the manuscript can be accpeted. Please, enhance the quality of the figures.